# Assessment of Hot Corrosion in Molten Na_2_SO_4_ and V_2_O_5_ of Inconel 625 Fabricated by Selective Laser Melting versus Conventional Technology

**DOI:** 10.3390/ma15124082

**Published:** 2022-06-08

**Authors:** Teodor Adrian Badea, Dan Batalu, Nicolae Constantin, Alexandru Paraschiv, Delia Pătroi, Laurentiu Constantin Ceatra

**Affiliations:** 1Romanian Research and Development Institute for Gas Turbines COMOTI, 220D Iuliu Maniu Av., 061126 Bucharest, Romania; teodor.badea@comoti.ro (T.A.B.); alexandru.paraschiv@comoti.ro (A.P.); laurentiu.ceatra@comoti.ro (L.C.C.); 2Materials Science and Engineering Faculty, University Politehnica of Bucharest, 060042 Bucharest, Romania; nctin2014@yahoo.com; 3R&D National Institute for Electrical Engineering ICPE-CA Bucharest, 313 Splaiul Unirii, 030138 Bucharest, Romania

**Keywords:** Inconel 625, selective laser melting, hot corrosion, sodium sulphate (Na_2_SO_4_), vanadium pentoxide (V_2_O_5_)

## Abstract

Inconel 625 samples, obtained by Selective Laser Melting (SLM) and conventional technology, were tested for hot corrosion resistance against a molten mixture of Na_2_SO_4_ and V_2_O_5_. The assessments were performed in air, at 900 °C with exposure time of up to 96 h, and at 1000 °C for 8 h. Weight gain was higher for samples obtained by SLM, with 37.4% after 8 h, 3.98% after 24 h, 4.46% after 48 h, and 5.8% after 96 h at 900 °C (22.6% at 1000 °C, 8 h). Three stages of corrosion were observed, the first and last with a high corrosion rate, while the second one showed a slower corrosion rate. Corrosion behaviour depends on the morphology of the grain boundary, which can influence the infiltration of corrosive salts, and on the formation of Cr_2_NiO_4_ compound, which acts as a temporary barrier.

## 1. Introduction

Superalloys are widely used because of their resistance to high temperatures, corrosion, and oxidation, making them indispensable for the aerospace industry, the chemical industry, power plants, and other special applications. Although Inconel was developed in 1932 and Inconel 625 in the middle of the 1960s [1], there are many challenges related to their resistance when working in corrosive environments. The main challenges are regarding (1) the resistance of the superalloy to oxidation, and (2) the corrosion at high temperatures [2,3,4,5,6,7,8,9,10,11].

Inconel 625 is the superalloy used most often due to its excellent combination of yield, creep, and fatigue strength. Additionally, it does not always require precipitation heat treatment process. This nickel-based superalloy has a high chromium and molybdenum content, and niobium and molybdenum are distributed in the nickel-chromium FCC matrix [12,13].

Additive manufacturing (AM) applied to Inconel 625 is a technology perfectly suited to solving already-known problems of the material when complex shapes are needed, and if we consider its poor machinability, high hardness, and low thermal conductivity [14,15]. From the very beginning, AM adopted Inconel 625 as raw powder for technologies based on laser, plasma arc or electron beam as the energy source [6,16,17,18]. High energy AM technologies can be divided into two categories [19]: (1) powder-bed fusion (PBF, including Selective Laser Melting, SLM), and (2) directed energy deposition (DED).

SLM technology uses a high energy laser beam, introducing a very high cooling rate to the process and subsequently affecting the microstructure [20,21,22,23,24]. SLM provides a unique and completely different microstructure compared to the equilibrium structure of conventionally fabricated materials. Due to the high cooling rate, grains grow mostly perpendicular to the substrate and elongated, columnar grains are formed [22,23,24]. This particularity produces a build direction anisotropy, with textured grains.

In the 1940s, hot corrosion provided solid proofs to be considered a serious problem due to the degradation of boiler tubes from steam generating plants. Since then, hot corrosion had been closely observed until the late 1960s, when it became a serious problem due to corrosive attacks on the engine parts of helicopters that were used in the Vietnam War and flew over or near the sea [10,25].

The hot corrosion process is based on the degradation of the material when it interacts at high temperatures with Na, V, S, and Cl from ash, fuel residues or other corrosive environments.

Hot corrosion is of two major types: type 1, the high-temperature hot corrosion (HTHC) above 750 °C, and type 2, low-temperature hot corrosion (LTHC) in the temperature range of 550–750 °C. In HTHC, Na_2_SO_4_ can reach the melting point at 884 °C, while in the LTHC it can form a Na_2_SO_4_-MSO_4_ eutectic (M = Ni, Co, and Fe) [26,27,28].

Problems occur when complex mixtures of molten sodium sulphate (Na_2_SO_4_) and vanadium pentoxide (V_2_O_5_) enter the power generation systems due to the poor fuel quality [29,30]. Na_2_SO_4_ is carried into the turbine inlet along with the air or it can be produced by a reaction between sodium chloride (NaCl) and sulphur impurities from the fuel [31]. During combustion, the vanadium porphyrin present in the fuel is converted into V_2_O_5_. The reaction between Na_2_SO_4_ and V_2_O_5_ forms new compounds with eutectic temperature below 600 °C. The eutectic compounds will easily melt and deposit on the turbine blades, causing corrosion with dramatic consequences [32,33].

Alloying elements have a major impact on hot corrosion resistance of superalloys. Inconel 625 includes elements well known for improving the corrosion resistance, namely Cr and Mo. Over 15 wt.% of Cr improves the hot corrosion performance, especially HTHC [34,35]. This is due to the formation of the adherent layer of Cr_2_O_3_ which maintains the acidity/basicity of the salt in an optimum range [36,37]. Mo is added as a solid solution strengthening and carbide former (Mo_6_C, MoC and Mo_23_C_6_) [38], but has a negative impact over corrosion resistance due to its interaction with Na_2_SO_4_ (Equation (1)), when it forms an acid salt, increasing the corrosion rate [39,40,41].
Na_2_SO_4_ (liq) + MoO_3_ (liq) = Na_2_MoO_4_ (liq) + SO_3_ (g)(1)

In our work, we assessed the behaviour at hot corrosion of Inconel 625, obtained either by additive manufacturing or by conventional technology, and compared the results. The corrosion agent was a mixture of Na_2_SO_4_ and V_2_O_5_ powders.

## 2. Materials and Methods

Two commercial raw materials of Inconel 625 were used for our study, one as annealed rolled plate (Bibus Metals, Păulești, Romania) and the other as powder for additive manufacturing (AM) (UK 81572 lot, LPW Technology, Ltd., Runcorn, UK, 15–45 μm particle size, and chemical composition from Table 1). The chemical composition of AP was determined by optical emission spectrometry (OES) with a WAS PMI-MASTER PLUS spectrometer (Table 1).

Na_2_SO_4_ and V_2_O_5_ with purities of 99% and 99.6%, respectively, were acquired from Carlo Erba Reagents SAS.

Samples of 20 mm × 15 mm × 4 mm were cut from 3D printed plates (denoted as 3DPP). Printed samples were obtained by SLM additive manufacturing technique, with a LASERTEC 30 SLM printer, at an angle of 90°. The following SLM parameters were used: 250 W laser power, 750 mm/s laser speed, 50 μm layer thickness, and 0.11 mm hatch distance.

The weight of each sample was determined with a Cole-Parmer Ohaus PX224 Pioneer analytical balance (accuracy 10^−4^ g). The apparent density of the 3DPP samples was determined by the Archimedes method in distilled water.

After printing, the 3DPP samples were annealed at 1000 °C for 1 h, air cooled, and then polished on SiC abrasive paper, up to #1200, prior to hot corrosion tests. Annealing was performed for stress release and homogenization, and to obtain similar start conditions as AP sample.

Optical microscopy (OM) was performed on a Axio Vert. A1 MAT Optical microscope (Carl Zeiss Instruments, București, Romania). The polished surface of the samples was etched by immersion in aqua regia for 10–15 s.

For hot corrosion tests, a mixture of 50 wt.% Na_2_SO_4_ and 50 wt.% V_2_O_5_ powders were spread on the polished surface of the samples. Previously, the surface of the samples were cleansed with water, ultrasonicated in acetone, and dried. The area density of the mixture was ~5 mg/cm^2^. The powder mixture was evenly spread on the surface of the samples with a brush. The prepared samples were placed in alumina crucibles and heated in a Nabertherm LH 30/14 furnace. Two different heat treatment (HT) temperatures were proposed, the first one at 900 °C and the second one at 1000 °C. The heating rate was 250 °C/h, from room temperature to 900/1000 °C. The selected holding times for 900 °C HT were 8, 24, 48, and 96 h, while for 1000 °C it was 8 h. Cooling was achieved in the furnace.

The microstructural and surface morphology analysis of the corroded specimens were investigated by scanning electron microscopy (SEM), using F50 Inspect SEM equipped with an energy-dispersive X-ray spectrometer (EDS) EDAX APEX 2i, SDD Apollo X detector (FEI Company, Brno, Czech Republic), and EDAX Genesis software v6.29 (EDAX Inc. Ametek MAD, Mahwah, NJ, USA). Backscattered electron images with a low voltage high contrast detector (vCD detector, FEI Company) were captured to highlight the chemical composition differences at the interface between the oxide layer and the substrate. The element distribution maps and micro-compositional analysis on two micro-areas and element maps were performed by SEM-EDS on the vCD images obtained on the cross-section of the specimens, using an acceleration voltage of 30 kV, a take-off angle of 35.6°, a spot size of 4 nm, and a working distance of 11 mm.

XRD measurements were made with a theta—2theta configured D8 Discover (Bruker AXS, Karlsruhe, Germany) by using CuK_α_ radiation (λ = 1.540598 Å). X-ray diffraction patterns were acquired using a 1D LYNXEYE detector at an angular increment of 0.04° and a 10 s/step scan speed. A zero-diffraction background of Si plate (Siltronix, Archamps, France) was used. The XRD patterns were indexed using the ICDD PDF 2 database and semi-quantitatively evaluation was made by Rietveld analysis using TOPAS software (Bruker, Karlsruhe, Germany).

## 3. Results and Discussion

The measured chemical compositions of the raw materials are similar (Table 1).

The measured apparent density of the 3DPP sample was 99.70%, indicating a dense sample with low porosity (0.3%).

Weight gain before and after corrosion tests at 900/1000 °C and 8/24/48/96 h was measured (Table 2) as the difference between the final and initial weight. A similar trend of samples was noticed, with a slightly higher gain for the 3DPP sample (Figure 1). Weight gain is higher for samples obtained by SLM, with 37.4% after 8 h, 3.98% after 24 h, 4.46% after 48 h, and 5.8% after 96 h at 900 °C (22.6% at 1000 °C after 8 h). The 3DPP samples have considerably more defects (porosity, weak grain boundaries between melted strings, etc.) than the AP sample, hence the corrosion rate is expected to be higher, but it is remarkable that the differences are not significant and the hot corrosion behaviour is similar. The hot corrosion test at 1000 °C/8 h was performed to show that, at over 900 °C, the corrosion becomes much more aggressive. The weight gain is much higher at 1000 °C/8 h than at 900 °C/48 h.

The higher corrosion rate at 1000 °C is due to the salt melting and the formation of basic and acidic fluxes on the surface of samples [42]. Additionally, the reactions are intensified by the migrating metal ions from the sample and by chemical reactions with oxidizing agents from the molten salt [43]. Three stages of weight gain can be observed at 900 °C (Figure 1). In the first stage (<24 h), reaction products formed on the surface of the sample prevent the formation of protective layers, leading to a higher rate of corrosion. However, in the second stage, new protective phases are formed, slowing down the corrosion process (24–48 h, [44]). For longer hot corrosion exposure times (>48 h), the protective barrier is penetrated and the corrosion rate increases again.

Kolta et al. [45] studied the kinetics of reactions between Na_2_SO_4_ and V_2_O_5_ and revealed that the reaction rate depends on both the temperature (between 600–1300 °C) and their molar ratios. Vanadates have a relatively low melting point that starts at 535 °C, and other metal oxides dissolved in vanadates can lower this temperature even more. Kolta et al. reported that the resulting slags from a diesel engine which has a predominant concentration of sodium sulphate and sodium vanadate have a melting point as low as 400 °C.

Superalloys usually undergo two stages of the hot corrosion process: an initiation stage and a propagation stage. All corrosion-resistant alloys degrade through these two stages, and this results in using selective oxidation to grow resistance to oxidation or corrosion [41,46].

In the case of hot corrosion tests above 890 °C, the mixture of corrosive salts Na_2_SO_4_ + V_2_O_5_ melts due to the low melting temperatures of Na_2_SO_4_ (884 °C) and V_2_O_5_ (681 °C), resulting in acidic and basic slag over the sample surface. In the first stage of corrosion, the proposed reactions are [47,48]:Na_2_SO_4_ (liq) → Na_2_O (liq) + SO_3_ (g)(2)
Na_2_O (liq) + V_2_O_5_ (liq) → 2NaVO_3_ (liq)(3)

In this stage, chromium oxide is the only one compound that meets the characteristics necessary to prevent the diffusion of vanadium, sulphur, and oxygen [49,50]. A stable chromium oxide phase (Cr_2_O_3_) forms after the consumption of carbide phases in the reaction with the oxygen from the air, at high temperature [43,51], according to the following sequence:Cr_3_C_2_ → Cr_7_C_3_ → Cr_23_C_6_ → Cr_2_O_3_(4)

Then, in the second stage of the corrosion, two simultaneous reactions occur. Cr_2_O_3_ interacts with NiO and NaVO_3_ (Equations (5) and (6)). The resulted Na_2_O reacts with V_2_O_5_ (Equations (7) and (8)) and continues the corrosive cycle [52,53]:Cr_2_O_3_ + NiO → NiCr_2_O_4_(5)
Cr_2_O_3_ + 2NaVO_3_ → 2CrVO_4_ + Na_2_O(6)
Na_2_O (liq) + V_2_O_5_ (liq) → 2NaVO_3_ (liq)(7)
3Na_2_O (liq) + V_2_O_5_ (liq) → 2Na_3_VO_4_(8)

The optical analysis reveals the microstructures of the annealed AP and 3DPP samples. The AP microstructure (Figure 2a) shows uneven polygonal equiaxed grains and annealing twins. The annealed 3DPP shows two interwoven microstructures. The background microstructure shows layered melted strings, either in longitudinal section, or transversal section (Figure 2b—inset, before annealing). Another scribbled-like microstructure can be noticed after annealing and etching (Figure 2b), and it corresponds to the new crystallites (partial recrystallization) formed after annealing at 1000 °C/1 h. The new columnar grains are elongated along a perpendicular direction on the printed layers (Figure 2b).

After hot corrosion tests, a powder with a new chemical composition formed on the surface of the samples and was analysed by XRD (Figure 3). The results show the formation of four new oxides: NiO (ICDD PDF2 file #00-044-1159), Cr_2_NiO_4_ (ICDD PDF2 file #03-065-3150), NiMoO_4_ (ICDD PDF2 file #00-033-0948), and FeNbO_4_ (ICDD PDF2 file #00-016-0357).

Patterns for 900 °C/96 h and 1000 °C/8 h are similar (Figure 3a,b), but ratios of the oxides are different based on peak intensity and Rietveld calculation (Table 3). Rietveld refinement was performed by intensity and profile fitting. Reliability factors of the weighted profile factor R_wp_ being less than 10, it should be considered rather a semi-quantitative analysis.

SEM and EDS investigations (Figure 4, Figure 5, Figure 6 and Figure 7) were made on the transversal sections of the samples to observe the sample-layer interface and the elements distribution in sample and layer. A thicker layer can be observed on samples tested at 1000 °C for 8 h, almost double if compared with samples tested at 900 °C for 96 h. Some cracks can be noticed in the layers of samples tested at 900 °C and large pores in the layers of samples tested at 1000 °C. A more compact layer forms on the surface of the samples, which is more evident in samples tested at 1000 °C (Figure 6 and Figure 7). Diffusion of S, Na, and V ions occurred in all metallic samples. A uniform distribution of Na and V can be noticed for 3DPP at 1000 °C (Figure 7), while in the other cases (Ni, Cr, Mo, Fe, etc.), a gap between layer and sample can be observed (Figure 6). A diffusion of elements from metallic samples to corrosion layer also occurred. Ni, Cr, Mo, Fe and Nb were identified on the side corresponding to the corrosion layer because of their diffusion. Oxygen is present both in the corrosion layer and in the sample, with a much richer presence in the layer; it has two sources, corrosion powder and the air. This promotes the idea that complex and aggressive diffusion and reactions can occur between the corrosive layer and sample, both in AP and 3DPP.

A quantitative analysis was performed on corrosive layers. Two points for investigations were selected, the first one (1) closer to the layer surface and the second one (2) closer to the substrate (Figure 4, Figure 5, Figure 6 and Figure 7, Table 4). The results show a clear difference of chemical composition between the two selected areas, hence there is a chemical composition gradient and a possibility that there are more phases present than those identified by XRD analysis, although in a low amount, such as Cr_2_O_3_ [48].

Both areas are similar in terms of type of elements, but different in terms of their concentration. For example, the intermediate area (2) is richer in oxygen (AP-2 at 900 °C/96 h) and (3DPP-2 at 1000 °C/8 h). It is not the same case for 3DPP-2 (900 °C/96 h) and AP-2 (1000 °C/8 h). A higher diffusion of Ni and Cr from the sample to corrosive layer for the 3D printed sample can be noticed within the same parameters of hot corrosion tests (Table 4). This is not the case for Mo, which has a lower diffusion in 3DPP than in AP. Based on the ratio, Mo is the third element present in the corrosive layer, after Ni and Cr. Mo can interact with testing salts and cause induced acidic fluxing, giving rise to accelerated spalling and sputtering of the corrosion scale in alloy [39,40,41]. The difference between 3DPP and AP corrosion behavior could be explained by the main degradation mechanism that will be discussed below (some exceptions may occur due to microanalysis limitations and nonuniformity of chemical composition in layers).

This diffusion is expected to occur mainly at the surface of the samples and along the grain boundaries, and to promote the formation of micro-cracks, evidenced after etching with aqua regia. The degradation through corrosion at grain boundaries can be best observed at samples heated at 1000 °C after etching with aqua regia (Figure 8). The reagent easily dissolved a significant number of grain boundaries where corrosive products infiltrated.

A possible degradation mechanism that can influence the corrosion rate of the material is based on the quantity of NiCr_2_O_4_ spinel phase, formed on the surface of samples and on chromium and nickel oxides. The NiCr_2_O_4_ spinel phase is prone to form through reaction of Cr_2_O_3_ with NiO, which subsequently prevents the diffusion of oxygen in the coating, thus reducing the corrosion rate [36]. Although NiCr_2_O_4_ is present in a larger amount in AP than in 3DPP, AP has a slightly lower weight gain during the hot corrosion process.

The main mechanism that could explain the corrosion differences between samples is related to the morphology of the grain boundaries. If we consider the grain boundary corrosion, a fine and compact structure can slow down the corrosion process [54] by spreading the same amount of corrosive salts over a larger surface. In our case, less, deeper, and larger grain boundaries can be observed on 3DPP samples (Figure 8c,d). This can be explained based on a different morphology of grain boundaries in the 3DPP samples, with more defects, such as pores, that can allow a higher corrosion rate. This behaviour requires post-processing of 3DPP samples to prevent the corrosion, for example, the deposition of corrosion-resistant layers designed for different types of corrosive agents. This infiltration of corrosive salts can be corelated with the amount of protective phase NiCr_2_O_4_. Nickel chromite reduces the aggressiveness of the salts, allowing the base material to form a larger amount of protective phase.

## 4. Conclusions

Hot corrosion resistance behaviour of Inconel 625, fabricated by conventional technology (cold rolled and annealed) and by SLM, a nonconventional technology based on micro-welding of thin strings, was assessed at 900 °C and 1000 °C at different holding times (8, 24, 48, and 96 h). Based on experimental results at 900 °C, we can conclude that the corrosive rate is faster during the first 24 h, followed by a slowing down in the next 24 h, and an increase again after that. Samples obtained by SLM have a slightly higher mass gain during corrosion tests than the those obtained by a conventional route. This behaviour is expected, as the samples have some inherent defects specific to SLM technology, such as porosity.

Four types of oxides were identified by XRD in the corrosive layers (NiO, Cr_2_NiO_4,_ NiMoO_4_, and FeNbO_4_), indicating the diffusion of metallic ions from samples. This also proves that the selected salts for corrosion tests, which form during exploitation of engines, can affect the integrity and service life. The formation of the Cr_2_NiO_4_ spinel can have the advantage of slowing down the corrosion rate.

We consider that at least two degradation mechanisms that influence the corrosion rate of the Inconel 625 are involved and interconnected. One mechanism is based on chemical reactions at high temperatures between deposited salts and the alloy, with the diffusion of metallic ions to the deposited layer. The second mechanism is related to the different morphologies of the grains from samples obtained by conventional technology and by SLM.

Weight gain was higher for samples obtained by SLM, with 37.4% after 8 h, 3.98% after 24 h, 4.46% after 48 h, and 5.8% after 96 h, at 900 °C (22.6% at 1000 °C, 8 h).

## Figures and Tables

**Figure 1 materials-15-04082-f001:**
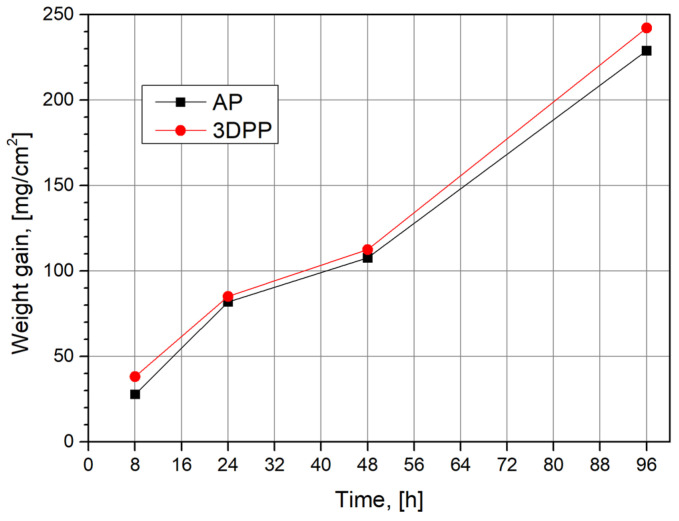
Weight gain vs. time for AP and 3DPP samples at 900 °C.

**Figure 2 materials-15-04082-f002:**
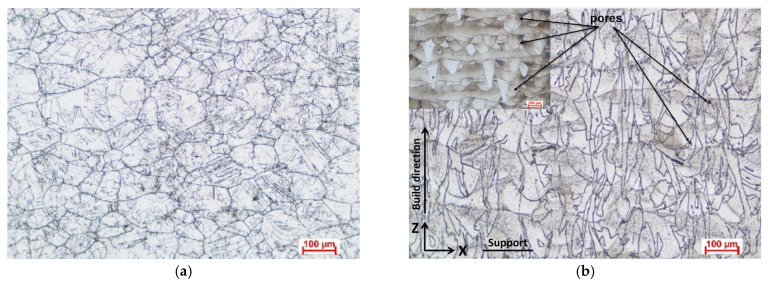
The microstructure of Inconel 625 for annealed AP (**a**) and 3DPP (**b**).

**Figure 3 materials-15-04082-f003:**
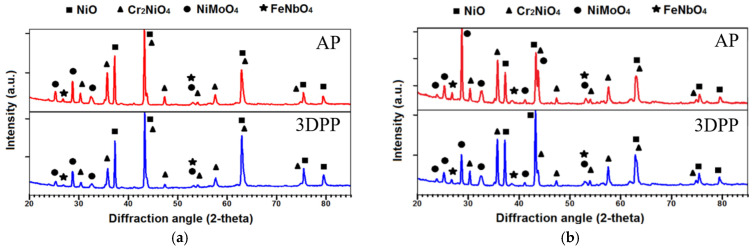
XRD patterns of the corrosive layer after hot corrosion at 900 °C for 96 h (**a**) and at 1000 °C for 8 h (**b**).

**Figure 4 materials-15-04082-f004:**
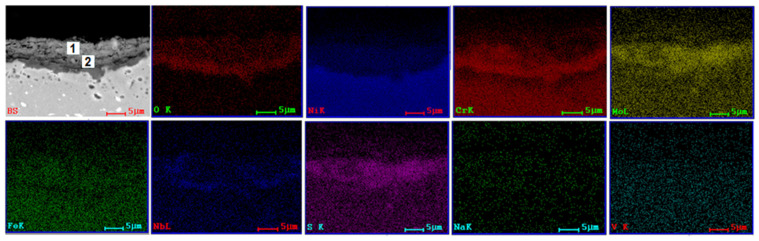
SEM image (backscattering mode, BS) and EDS distribution maps of the elements corresponding to SEM image for cross section of AP (900 °C, 96 h).

**Figure 5 materials-15-04082-f005:**
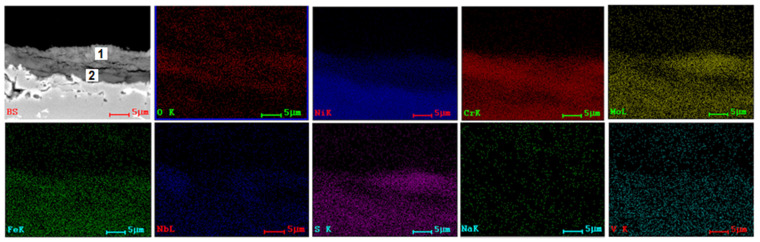
SEM image (BS) and EDS distribution maps of the elements corresponding to SEM image for cross section of 3DPP (900 °C, 96 h).

**Figure 6 materials-15-04082-f006:**
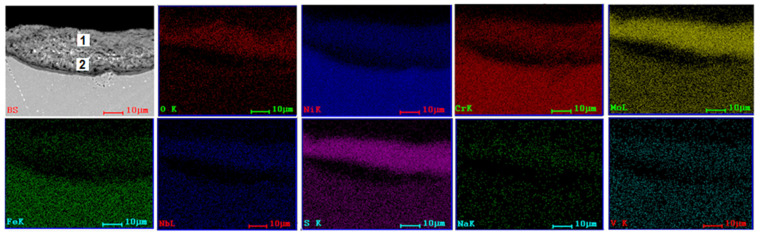
SEM image (BS) and EDS distribution maps of the elements corresponding to SEM image for cross section of AP (1000 °C, 8 h).

**Figure 7 materials-15-04082-f007:**
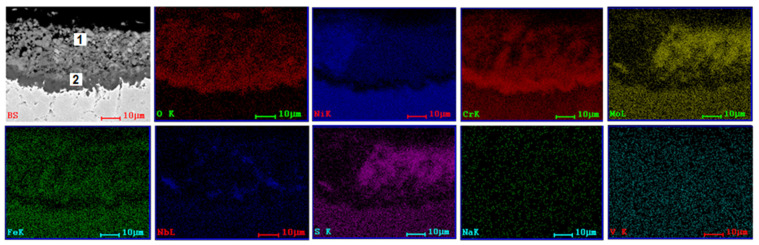
SEM image (BS) and EDS distribution maps of the elements corresponding to SEM image for cross section of 3DPP (1000 °C, 8 h).

**Figure 8 materials-15-04082-f008:**
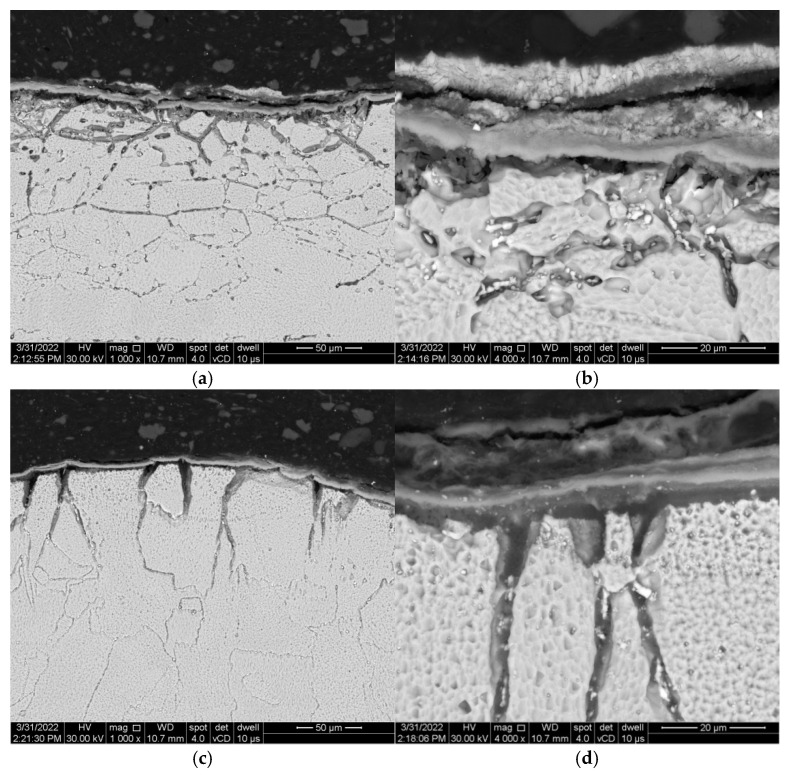
SEM images with micro-cracks formed at grain boundaries in AP (**a**,**b**) and 3DPP (**c**,**d**) at 1000 °C/8 h, revealed by etching with aqua regia.

**Table 1 materials-15-04082-t001:** Chemical composition of AP plate and 3DPP powder (wt.%).

Elements	Ni	Cr	Mo	Fe	Nb	Co	Mn	Al	Ti	Si	C
AP	61.40	21.40	8.19	4.45	3.40	0.11	0.36	0.11	0.22	0.28	0.06
AM powder	62.26	20.70	8.90	4.10	3.77	0.10	0.01	0.06	0.07	0.01	0.02

**Table 2 materials-15-04082-t002:** Weight gain of specimens after hot corrosion tests (±SD).

Sample	Weight Gain, [mg/cm^2^]				
	900 °C/8 h	900 °C/24 h	900 °C/48 h	900 °C/96 h	1000 °C/8 h
AP	27.8 ± 0.2	81.83 ± 0.01	107.68 ± 0.12	228.9 ± 0.02	163.2 ± 0.14
3DPP	38.2 ± 0.12	85.09 ± 0.12	112.49 ± 0.04	242.2 ± 0.12	200.1 ± 0.02

**Table 3 materials-15-04082-t003:** Oxides formed in the corrosive layers and their ratios (wt.%).

Samples’ Surface Oxides	NiO	Cr_2_NiO_4_	NiMoO_4_	FeNbO_4_
AP (900 °C/96 h)	44.2	20.3	33.8	1.7
AP (1000 °C/8 h)	29.3	35.5	32.4	2.8
3DPP (900 °C/96 h)	53.6	16.3	28.6	1.5
3DPP (1000 °C/8 h)	38.7	24.5	34.2	2.6

**Table 4 materials-15-04082-t004:** Chemical composition measured by EDS in two areas from corrosive layer.

Sample-Area	Chemical Composition (wt.%)								
	O	Ni	Cr	Mo	Fe	Nb	S	Na	V
AP-1 (900 °C/96 h)	18.31	23.54	10.90	26.11	2.68	4.52	11.86	1.16	0.93
AP-2 (900 °C/96 h)	24.74	22.07	20.47	15.20	4.13	4.45	6.82	1.30	0.81
3DPP-1 (900 °C/96 h)	18.92	25.11	13.80	22.75	2.83	3.88	10.42	1.21	1.08
3DPP-2 (900 °C/96 h)	13.26	26.87	27.72	12.19	3.28	9.32	5.15	0.93	1.28
AP-1 (1000 °C/8 h)	23.69	18.29	7.03	28.98	1.62	4.47	13.36	1.69	0.86
AP-2 (1000 °C/8 h)	15.27	39.01	10.58	18.81	2.39	2.99	8.54	1.23	1.16
3DPP-1 (1000 °C/8 h)	18.70	23.20	14.52	22.19	2.53	6.97	9.81	1.14	0.94
3DPP-2 (1000 °C/8 h)	26.75	12.11	46.01	3.95	2.49	4.99	1.61	1.08	1.02

## Data Availability

The authors declare that, to the best of their knowledge, all data and material comply with field standards. Data are available by request.

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
