# Peer review of "Assessment of Hot Corrosion in Molten Na2SO4 and V2O5 of Inconel 625 Fabricated by Selective Laser Melting versus Conventional Technology"

_materials, 2022, doi:10.3390/ma15124082_

Round 1

Reviewer 1 Report

This work made a comparative study on the high-temperature corrosion test between the laser additively manufacturing and annealed (maybe from casting) Inconel 625 superalloys.

However, a major revision is recommended.

Please provide the reason for the annealing for the SLM sample before high-temperature corrosion tests. The AP sample is re-annealed, again?

The authors concluded the internal defects is the major reason for the higher weight gain during corrosion. the related mechanism analysis is not given. the SEM observation on the pore defects should be given in the analysis.

Most importantly, the profound discussion is not made on the different corrosion mechanism, especially concerning the corrosive layer. The readers are wondering about whether the laser additively manufactured Inconel 625 superalloys have different high-temperature corrosion behaviors, as well as the mechanism. Does the microstructure features in laser additive manufacturing superalloy attribute to the corrosion process? Even though the SLM sample was annealed before corrosion test, the microstructure still is more refined.

At last, the grammar mistakes and typos should be modified. The format of the figures should be improved as well, for example, the Fig. 8.

Reviewer 2 Report

Comment 1: Title should be revised and improved.

Comment 2: Qualitative informations are missing in abstract. Abstract should be concise and the authors need to improve with more specific short results.

Comment 3: The purity of used products should be mentioned in materials and method section.

Comment 4: In all manuscript "hours" should be revised as "hrs".

Comment 5: Report the standard deviation and errors bars in Table 2. The following references should added.

Chemical Data Collections 31 (2021) 100619 (https://doi.org/10.1016/j.cdc.2020.100619),

Chemical Physics Letters 783 (2021) 139081 (https://doi.org/10.1016/j.cplett.2021.139081).

Comment 6: Report the scattering bars in Figure 1. The following references should added.

Construction and Building Materials. 270 (2021) 121454. (https://doi.org/10.1016/j.conbuildmat.2020.121454),

Journal of Molecular Liquids 337 (2021) 116492 (https://doi.org/10.1016/j.molliq.2021.116492).

Comment 7: The introduction section should be modified though citing recent references (2021 and 2022) related studies and indicating the novelty of the study compared to the carried works.

Comment 8: Compare your results with literature ones.

Comment 9: Conclusion is too, conclusion should be improved.

Comment 10: Level of English is good however in a few places some syntax errors are present. At some places two or more words joined together that should be corrected.

Reviewer 3 Report

The present research compares the hot corrosion behavior of IN 625 fabricated by SLM and conventional technology. The complex compounds which forms during hot corrosion process, were characterized by XRD and EDS analysis and finally, two degradation mechanisms that can influence the corrosion rate were proposed. It is recommended for publication after addressing the following issues:

1- The keywords are insufficient.

2- Line 66: The name of technique was missed (optical emission spectrometry).

3- Line 123: "The reaction are intensified by the …" is not discussed In Ref. [35]. The reference should be modified.

4- In Fig. 2 (b), indicate the building direction.

5- Line 211: Why Mo has a lower diffusion in 3DPP than in AP?

6- In Table 4, it is known that the diffusion coefficient of Mo is lower than that of Cr and Fe, why the Mo content is higher in the layer 1 than the layer 2?

7- A comparison between Fig. 8 (a) and (d) shows that the grain boundary density which involves in the hot corrosion process is higher in AP than 3DPP. Hence, the corrosion rate is higher in the AP specimen. I think that although, the corrosion cracks are deeper in the 3DPP specimen but they are not larger than the AP specimen. How can be explained the lower density of corrosion cracks in 3DPP than AP?

Round 2

Reviewer 3 Report

The comments were replied and it can be accepted after correcting the following minor issue.

Line 80: The name of technique was mentioned OES but in the title of Table 1, it was mentioned XRF. Which technique was utilized?

Author Response

Dear Reviewer,

Thank you for your important observation.

The technique used was OES. We deleted from the title of Table 1 "measured by XRF".